# *Maackia amurensis* Rupr. et Maxim.: Supercritical CO_2_ Extraction and Mass Spectrometric Characterization of Chemical Constituents

**DOI:** 10.3390/molecules28052026

**Published:** 2023-02-21

**Authors:** Mayya P. Razgonova, Elena I. Cherevach, Lyudmila A. Tekutyeva, Sergey A. Fedoreyev, Natalia P. Mishchenko, Darya V. Tarbeeva, Ekaterina N. Demidova, Nikita S. Kirilenko, Kirill Golokhvast

**Affiliations:** 1N.I. Vavilov All-Russian Institute of Plant Genetic Resources, B. Morskaya 42-44, 190000 Saint Petersburg, Russia; 2Department of Pharmacy and Pharmacology, School of Biomedicine, Far Eastern Federal University, Sukhanova 8, 690950 Vladivostok, Russia; 3G.B. Elyakov Pacific Institute of Bioorganic Chemistry, Far Eastern Branch of Russian Academy of Science, Prospect 100 let Vladivostoku 159, 690022 Vladivostok, Russia; 4Laboratory of Supercritical Fluid Research and Application in Agrobiotechnology, The National Research Tomsk State University, Lenin Str. 36, 634050 Tomsk, Russia; 5Siberian Federal Scientific Centre of Agrobiotechnology, Centralnaya, Presidium, 633501 Krasnoobsk, Russia

**Keywords:** *Maackia amurensis*, CO_2_ extraction, tandem mass spectrometry, polyphenols, bioactive compounds

## Abstract

Three types of extraction were used to obtain biologically active substances from the heartwood of *M. amurensis*: supercritical CO_2_ extraction, maceration with EtOH, and maceration with MeOH. The supercritical extraction method proved to be the most effective type of extraction, giving the highest yield of biologically active substances. Several experimental conditions were investigated in the pressure range of 50–400 bar, with 2% of ethanol as co-solvent in the liquid phase at a temperature in the range of 31–70 °C. The most effective extraction conditions are: pressure of 100 bar and a temperature of 55 °C for *M. amurensis* heartwood. The heartwood of *M. amurensis* contains various polyphenolic compounds and compounds of other chemical groups with valuable biological activity. Tandem mass spectrometry (HPLC-ESI—ion trap) was applied to detect target analytes. High-accuracy mass spectrometric data were recorded on an ion trap equipped with an ESI source in the modes of negative and positive ions. The four-stage ion separation mode was implemented. Sixty-six different biologically active components have been identified in *M. amurensis* extracts. Twenty-two polyphenols were identified for the first time in the genus *Maackia*.

## 1. Introduction

*Maackia amurensis* Rupr. et Maxim. is the only representative of the Fabaceae family in the flora of the Russian Far East. Probably, this species can be considered a relic of the Tertiary flora, which survived in more severe climatic conditions than the species of the genera Cladrastis and Sophora. The distribution area of *Maackia amurensis* is in the Amur River basin and in the south of Primorsky Krai, Russia. The natural reserves of this plant are large and actively self-renewal [1,2,3]. A close species to maackia is *M. amurensis* Rupr. et Maxim. var. *buergeri* (Maxim.) C.K. Schneeder (Figure 1). However, its chemical composition differs sharply from that of the Far Eastern species, *M. amurensis* [2,3,4,5]. Until 1985, the only data on the chemical composition of Russian maackia were data on alkaloids contained in the bark and green parts of the plant. In the subsequent detailed chemical study of alcoholic extracts of heartwood, it was shown that the main components of maackia are plant polyphenols [3,4,5].

These include isoflavones: genistein, daidzein, retusin, afromosin, formononetin, orobol, tectorigenin, 3-hydroxyvestiton, pterocarpans maakiain, medicarpin [5,6,7]. The peculiarity of *Maackia amurensis* growing in Primorye is the high content of monomeric stilbenes resveratrol and piceatannol and isoflavonstilben maackiasin in its wood [5,6]. These polyphenols were not found in the variety *Maackia amurensis* (var. buergeri) growing in Japan [1].

In addition to monomeric stilbenes and isoflavones, the polar fractions of *Maackia amurensis* extracts contain oligomeric stilbenes maackin, scirpusin A, scirpusin B, maackin A (XVII), and stilbenolignan maackolin [8,9].

The polyphenolic complex from *M. amurensis* heartwood, called Maksar^®^ preparation, is registered in the Russian Federation as a hepatoprotective medicine (P N003294/01). Maxar^®^ increases the body’s antioxidant system activity and reduces the lipid peroxidation level. Its application in clinical practice showed that this drug is effective for treating liver fatty dystrophy. It prevents the increase in total serum lipid content and the development of hyperlipoproteinemia in experimental animals. Maxar^®^ also possesses antithrombogenic, antiplatelet, and antitumor properties [10,11]. Recently, new research has been carried out which shows that stilbenolignan maackolin may be a good candidate as a SARS-CoV-2 Mpro inhibitor in vivo studies [12].

The use of supercritical fluids in food material applications and more broadly in the food industry began in the late 1960s and probably represents the most successful application of supercritical fluids to date. The “green technology” of supercritical CO_2_ extraction using high pressures is an excellent technique for obtaining natural thermolabile compounds. In addition, there are no residues of organic solvents in the products, which occurs with conventional extraction methods—conventional solvents can be toxic, for example, in the case of methanol and hexane. Easy removal of the solvent from the final product, a high selectivity, and the use of moderate temperatures in the extraction process are the main attractive factors of SFE, leading to a significant increase in research for applications in the food and pharmaceutical industries [13].

Chemical reactions that have made the greatest contribution to food technologies were enzyme-catalyzed reactions [14], hydrogenations designed to control particular trans isomers occurring in lipid mixtures [15], and hydrolysis conducted in the presence of enzymes or a medium such as subcritical water [16]. Considerable activity in producing fine particles for use in the pharmaceutical industry began in the 1990s. In the last three decades, focus on the development of technologies was displaced by the combination of SCF technologies in the food industry and the obtainment of bioactive agents from natural matrixes [17,18].

In this research, supercritical CO_2_ extraction, MeOH maceration, and EtOH maceration of the samples of *M. amurensis* were used to obtain an effective amount of biologically active substances. We used a tandem mass spectrometry to carry out a phytochemical study involving a detailed metabolomic analysis of *M. amurensis.* The bark of *M. amurensis* was collected during expedition work near Ussuri River, Primorsky Krai, Russia (N 42°36′10″ E 131°10′55″), during the period from 1 to 20 August 2022.

## 2. Results

Three samples of wood substance of *M. amurensis* were subjected to supercritical CO_2_ extraction under different extraction conditions. The applied supercritical pressures ranged from 150 to 400 bar, and the extraction temperature ranged from 31 to 65 °C. The co-solvent EtOH was used in an amount of 2% of the total amount of solvent. A pronounced extraction extremum is shown in the 3D graph (Figure 2). The best extraction conditions for M. amurensis (heartwood) were the following: pressure of 100 bar and temperature at 55 °C. The total yield of biologically active substances under these extraction conditions was 4.3 mg per 100 mg of supercritical CO_2_ extract. The structural identification of each compound was carried out on the basis of their accurate mass and MS/MS fragmentation via the HPLC–ESI–ion trap–MS/MS. A total of 66 compounds were identified in extracts of M. amurensis based on their accurate MS and fragment ions by searching online databases and the references.

The research data presented in Figure 2 show the presence of a confident maximum of supercritical CO_2_ extraction in the pressure range of 100 to 150 bar and temperature range from 50 °C to 55 °C. In this range, the highest yield of biologically active compounds from the plant matrix of *M. amurensis* is observed. It should be noted that during the experiment, the extraction time was also small—1 h; therefore, what can we say about the effectiveness of the applied method of supercritical CO_2_-extraction?

## 3. Discussion

The number of constituents tentatively identified via tandem mass spectrometry was 66 chemical compounds (54 compounds from the polyphenol group and 12 compounds from other chemical groups). All the identified polyphenols and compounds from other chemical groups, their molecular formulas, and MS/MS data for *M. amurense* are summarized in Table A1 (Appendix A). Polyphenols are represented by the following chemical groups: flavones, flavonols, flavan-3-ols, flavanones, stilbenes, hydroxycoumarins. For the first time, 22 polyphenols were identified in *M. amurense* heartwood. There are polyphenols: flavones biochanin-A, 7-hydroxy-6,4′-dimethoxyisoflavone, trihydroxy methoxyflavone, cirsimaritin, dihydroxy-dimethoxy(iso)flavone, myricetin, cirsiliol, wighteone, luteone, dihydroxy tetramethoxyflavanone, hydroxy hexamethoxyflavone, odoratin-*O*-hexoside, 6,4′-dimethoxyisoflavone-7-*O*-glucoside, genistein *C*-glucoside malonylated, calycosin-7-*O*-beta-D-glucoside-6″-*O*-malonate, chrysoeriol 8-*C*-glucoside malonylated, apigenin 7-*C*-glucosyldideoxyhexoside, flavanones methyl-liquiritigenin, liquiritigenin dimethyl ester, padmatin, etc.

Figure 3, Figure 4, Figure 5 and Figure 6 show examples of the decoding spectra (collision-induced dissociation (CID) spectrum) of the ion chromatogram obtained using tandem mass spectrometry. The CID spectrum in negative ion modes of isoflavone 3-hydroxyvestitol from *M. amurense* is shown in Figure 3.

[M+H]^−^ ion produced four fragment ions at *m/z* 299.13, *m/z* 283.13, *m/z* 227.24, and at *m/z* 177.08 (Figure 3). The fragment ion at *m/z* 283.06 produced three characteristic daughter ions at *m/z* 267.08, *m/z* 240.1, and *m/z* 150.2. The fragment ion at *m/z* 267.08 produced one characteristic ion at *m/z* 224.11. It was identified in the references in the extract from *M. amurense* [5,6,7]. The following is a list of polyphenols found in extracts of the wood substance of *M. amurense*; it should be noted separately that similar polyphenols are found in the Astragalus genus. The CID spectrum in positive ion modes of odoratin-*O*-hexoside from *M. amurense* is shown in Figure 4.

[M+H]^+^ ion produced five fragment ions at *m/z* 415.4, *m/z* 358.34, *m/z* 331.31, *m/z* 277.24, and at *m/z* 250.33 (Figure 4). The fragment ion with *m/z* 415.4 produced six characteristic daughter ions at *m/z* 331.37, *m/z* 303.31, *m/z* 261.33, *m/z* 206.19, *m/z* 176.96, and at *m/z* 149.18. The fragment ion at *m/z* 331.37 produced two characteristic daughter ions at *m/z* 261.99 and *m/z* 233.27; they were identified in extracts from *Astragali Radix* [19,20,21]. The CID spectrum in the positive ion mode of formononetin-7-*O*-glucoside-6″-O-malonate from *M. amurense* is shown in Figure 5.

[M–H]^+^ ion produced one fragment ion at *m/z* 269.18 (Figure 5). The fragment ion with *m/z* 269.18 produced four characteristic daughter ions at *m/z* 254.16, *m/z* 237.19, *m/z* 213.25, and at *m/z* 163.16. The fragment ion at *m/z* 254.16 formed three daughter ions with *m/z* 237.15, *m/z* 226.17, and *m/z* 181.24; they were identified in extracts from *Astragali Radix* [19,20,21]. The CID spectrum in the positive ion mode of calycosin-7-*O-*beta-D-glucoside-6″-*O*-malonate from *M. amurense* is shown in Figure 6.

[M+H]^+^ ion produced four fragment ions with *m/z* 285.17, *m/z* 387.34, *m/z* 354.34, and *m/z* 198.24 (Figure 6). The fragment ion with *m/z* 285.17 formed four daughter ions with *m/z* 167.14, *m/z* 257.36, *m/z* 229.2, and *m/z* 179.18; they were identified in extract from *Astragali Radix* [19,20,21].

It should also be noted that a detailed analysis of the presence of polyphenols and biologically active substances from other chemical groups showed the highest number of flavonoids at supercritical CO_2_ extraction at a pressure of 100 bar—43 compounds. Accordingly, with other types of extraction investigated in this study, such as maceration with ethanol (20 compounds), maceration with methanol (23 compounds), and supercritical CO_2_-extraction at a higher extraction pressure (19 compounds), the yield efficiency of biologically active substances is much lower. (Table 1).

Thus, it can be stated that as a result of the most detailed study using tandem mass spectrometry, new data on the content of biologically active substances in *M. amurensis* have been obtained.

*M. amurensis* extracts exhibited different DPPH scavenging effects compared to quercetin (Table 2). CO_2_ extract obtained at 100 bar possessed the most considerable activity compared to quercetin, which is mainly due to the high content of monomeric and dimeric stilbenes. EtOH extract from *M. amurensis* was the least active, because the main components of this extract were glycosides of isoflavones, which are rather weak antioxidants.

## 4. Materials and Methods

### 4.1. Materials

Wood substance of *M. amurensis* was collected during expedition work near Ussuri River, Primorsky Krai, Russia (N 42°36′10″ E 131°10′55″), during the period from 1 to 20 August 2022. All samples were morphologically authenticated according to the current standard of Russian Pharmacopeia [22].

### 4.2. Chemicals and Reagents

HPLC-grade acetonitrile was purchased from Fisher Scientific (Southborough, UK), MS-grade formic acid was from Sigma-Aldrich (Steinheim, Germany). Ultra-pure water was prepared from a SIEMENS ULTRA clear (SIEMENS water technologies, Munich, Germany), and all other chemicals were analytical grade.

### 4.3. Fractional Maceration

The fractional maceration technique was applied to obtain highly concentrated extracts [23]. From 500 g of the wood substance, 20 g of wood were randomly selected for maceration. The total amount of the extractant (ethyl alcohol of reagent grade) was divided into 3 parts, and the parts of plant were consistently infused with the first, second, and third parts. The solid–solvent ratio was 1:20. The infusion of each part of the extractant lasted 7 days at room temperature.

### 4.4. Extraction

SC-CO_2_ extraction was performed using the SFE-500 system (Thar SCF Waters, Milford, CT, USA) supercritical pressure extraction apparatus. System options include: co-solvent pump (Thar Waters P-50 High Pressure Pump), for extracting polar samples; CO_2_ flow meter (Siemens, Munich, Germany), to measure the amount of CO_2_ being supplied to the system; and multiple extraction vessels, to extract different sample sizes or to increase the throughput of the system. The flow rate was 10–25 mL/min for liquid CO_2_ and 1.00 mL/min for EtOH. Extraction samples of 100 g of wood substance of *M. amurensis* were used. The extraction time was counted after reaching the working pressure and equilibrium flow, and it was 60–90 min for each sample.

### 4.5. Liquid Chromatography

HPLC was performed using Shimadzu LC-20 Prominence HPLC (Shimadzu, Kyoto, Japan) equipped with a UV sensor and C18 silica reverse phase column (4.6 × 150 mm, particle size: 2.7 μm) to perform the separation of multicomponent mixtures. The gradient elution program with two mobile phases (A, deionized water; B, acetonitrile with formic acid 0.1% *v/v*) was as follows: 0–2 min, 0% B; 2–50 min, 0–100% B; control washing 50–60 min 100% B. The entire HPLC analysis was performed with a UV-vis detector SPD-20A (Shimadzu, Kyoto, Japan) at a wavelength of 230 nm for identification compounds; the temperature was 50 °C, and the total flow rate 0.25 mL min^−1^. The injection volume was 10  μL. Additionally, liquid chromatography was combined with a mass spectrometric ion trap to identify compounds.

### 4.6. Mass Spectrometry

MS analysis was performed on an ion trap amaZon SL (BRUKER DALTONIKS, Bremen, Germany) equipped with an ESI source in the negative ion mode. The optimized parameters were obtained as follows: ionization source temperature: 70 °C; gas flow: 4 L/min; nebulizer gas (atomizer): 7.3 psi; capillary voltage: 4500 V; end plate bend voltage: 1500 V; fragmentary: 280 V; and collision energy: 60 eV. An ion trap was used in the scan range *m/z* 100–1.700 for MS and MS/MS. The capture rate was one spectrum/s for MS and two spectrum/s for MS/MS. Data collection was controlled using Windows software for BRUKER DALTONIKS. All experiments were repeated three times. A four-stage ion separation mode (MS/MS mode) was implemented.

### 4.7. Antiradical Activity

We determined the DPPH (2,2-diphenyl-1-picrylhydrazyl) scavenging effect of extracts from *M. amurensis* heartwood. The extracts were added to DPPH solution in MeOH (10^−4^ M) at concentrations from 1 to 85 µg/mL. We kept the reacting mixture in the dark at room temperature for 20 min. Then, we measured the absorbance at 517 nm using a Shimadzu UV-1800 spectrophotometer (Shimadzu, Canby, OR, USA). We used Equation (1) to calculate the DPPH radical-scavenging effect (%):(1)$$\mathrm{DPPH}\;\mathrm{scavenging}\;\mathrm{effect},\;\%\;=\frac{A_{0}-A_{x}}{A_{0}}\times 100,$$
where *A*_0_ is the absorbance of DPPH solution without *M. amurensis* extracts (blank sample); *A_x_* is the absorbance of DPPH solution in the presence of different concentrations of extracts.

Quercetin was used as a reference compound. All experiments were performed in triplicate. The half maximal inhibitory concentrations (IC_50_) for extracts were calculated by plotting the DPPH scavenging effect (%) against the concentrations of *M. amurensis* extracts. IC_50_ values are given as the mean ± SEM.

## 5. Conclusions

Three types of extraction were used to obtain biologically active substances from the wood substance of *M. amurensis*: supercritical CO_2_ extraction, maceration with EtOH, and maceration with MeOH. The supercritical extraction method proved to be the most effective type of extraction, giving the highest yield of biologically active substances. Several experimental conditions were investigated in the pressure range of 50–400 bar, with the used volume of co-solvent ethanol being 2% in the liquid phase at a temperature in the range of 31–70 °C. The most effective extraction conditions are: pressure of 100 bar and temperature at 55 °C for the wood substance of M. amurensis. The wood of *M. amurensis* contains various polyphenolic compounds and compounds of other chemical groups with valuable biological activity. Tandem mass spectrometry (HPLC-ESI—ion trap) was applied to detect target analytes. High-accuracy mass spectrometric data were recorded on an ion trap amaZon SL BRUKER DALTONIKS equipped with an ESI source in the mode of negative and positive ions. The four-stage ion separation mode was implemented. Sixty-six different biologically active components have been identified in M. amurensis extracts. Twenty-two polyphenols were identified for the first time in the genus *Maackia*.

These data could support future research for the production of a variety of pharmaceutical products containing ultra-pure extracts of *M. amurensis*. The richness of various biologically active compounds, including compounds of the polyphenol group and compounds of other chemical groups (amino acids, Omega fatty acids, sterols, triterpenoids, etc.), provides great opportunities for the design of new nutritional and dietary supplements based on extracts from this *Maackia* genus.

## Figures and Tables

**Figure 1 molecules-28-02026-f001:**
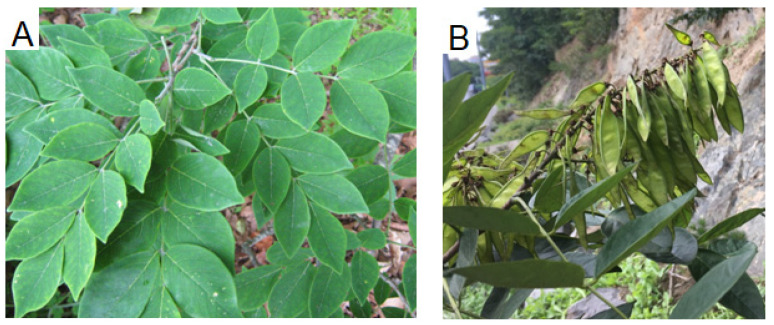
(**A**) *Maackia amurensis* L.; (**B**) fruits of *Maackia amurensis* species. Photo taken by N.P. Mishchenko (August 2020).

**Figure 2 molecules-28-02026-f002:**
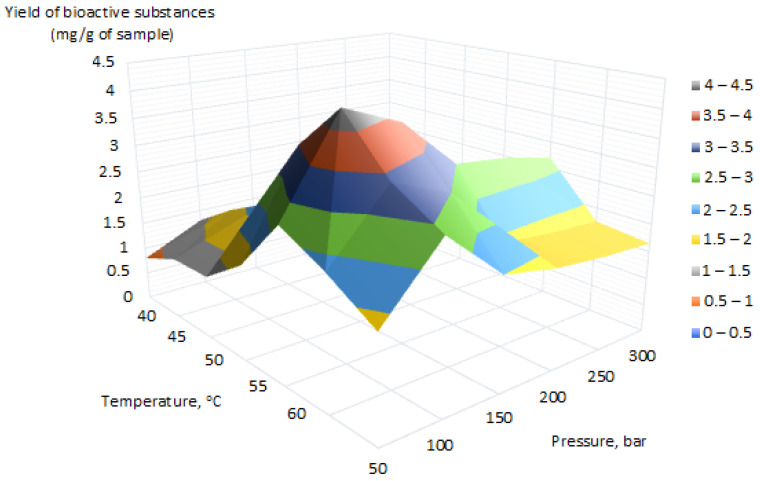
3D graph data of supercritical CO_2_ extraction. Complex yield of biologically active substances from CO_2_ extracts of wood substance of *M. amurensis*.

**Figure 3 molecules-28-02026-f003:**
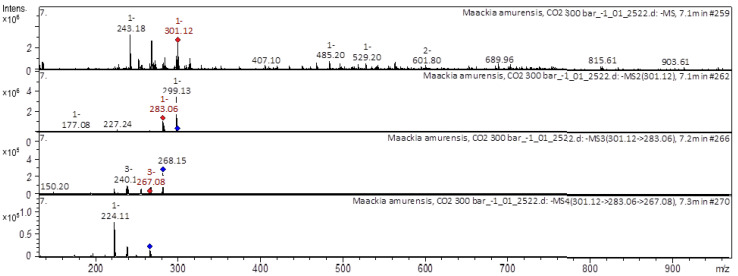
CID spectrum of 3-Hydroxyrvestitol from *M. amurense*, at *m/z* 275.01.

**Figure 4 molecules-28-02026-f004:**
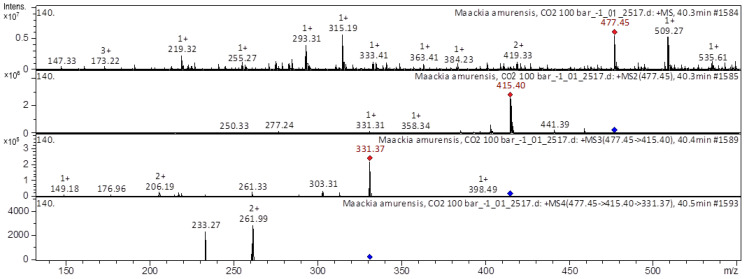
CID spectrum of odoratin-*O*-hexoside from *M. amurense*, at *m/z* 477.45.

**Figure 5 molecules-28-02026-f005:**
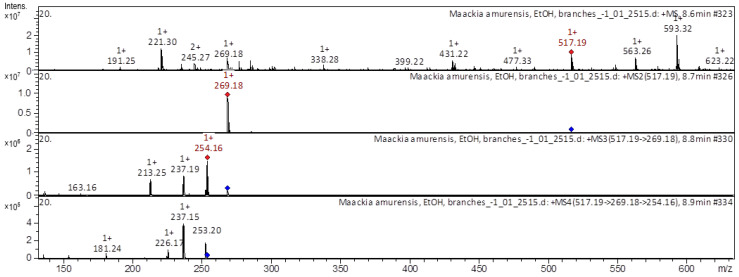
CID spectrum of formononetin-7-*O*-glucoside-6″-*O*-malonate from *M. amurense*, at *m/z* 517.19.

**Figure 6 molecules-28-02026-f006:**
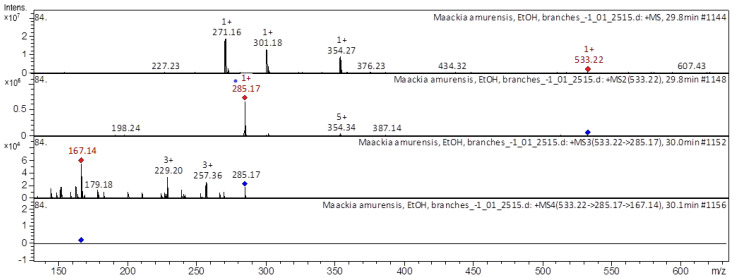
CID spectrum of calycosin-7-*O*-beta-D-glucoside-6″-O-malonate acid from *M. amurense*, at *m/z* 533.22.

**Table 1 molecules-28-02026-t001:** Identified chemical constituents by tandem mass spectrometry in three types of extraction: supercritical CO_2_ extraction, maceration with EtOH, maceration with MeOH.

№	Class of Compounds	Identified Compounds	MeOH Extraction	EtOH Extraction	Static CO_2_ Extraction	CO_2_ Extraction 100 Bar	CO_2_ Extraction 300 Bar
		POLYPHENOLS					
1	Flavone	2′-Hydroxyformononetin [Xenognosin B]	+				
2	Flavone	Daidzein [4′,7-Dihydroxyisoflavone; Daidzeol]	+			+	+
3	Flavone	Formononetin [Biochanin B; Formononetol]	+	+	+		
4	Flavone	Apigenin	+	+		+	+
5	Flavone	Genistein	+				+
6	Flavone	Calycosin [3′-Hydroxyformononetin]		+			+
7	Flavone	5-Methoxydaidzein	+	+	+	+	+
8	Flavone	Biochanin-A *		+			+
9	Flavone	Orobol	+		+	+	+
10	Flavone	7-Hydroxy-6,4′-dimethoxyisoflavone*				+	
11	Flavone	Afromosin [Castanin]			+	+	+
12	Flavone	Tectorigenin	+	+	+	+	+
13	Flavone	Trihydroxy methoxyflavone *				+	+
14	Flavone	3-Hydroxyvestiton				+	+
15	Flavone	Odoratin	+	+	+	+	+
16	Flavone	Cirsimaritin *				+	
17	Flavone	Dihydroxy-dimethoxy(iso)flavone *		+			
18	Flavone	Myricetin *				+	
19	Flavone	Cirsiliol *			+		
20	Flavone	Wighteone [Erythrinin B] *	+	+		+	+
21	Isoflavone	Luteone *		+			
22	Flavone	Dihydroxy tetramethoxyflavanone *	+				
23	Flavone	Hydroxy hexamethoxyflavone *				+	
24	Flavone	Formononetin (Glycosylated and methylated)				+	
25	Flavone	Odoratin-*O*-hexoside *	+			+	
26	Flavone	6,4′-Dimethoxyisoflavone-7-*O*-glucoside *				+	
27	Flavone	Formononetin-7-*O*-glucoside-6″-*O*-malonate		+			
28	Flavone	Genistein *C-*glucoside malonylated *	+				
29	Flavone	Maackiasin					+
30	Flavone	Calycosin-7-*O*-beta-D-glucoside-6″-*O*-malonate *		+			
31	Flavone	Chrysoeriol *8-C*-glucoside malonylated *					+
32	Flavone	Apigenin 7-C-glucosyldideoxyhexoside *		+			
33	Isoflavone	Ononin derivative		+			
34	Flavone	Apigenin 6-*C*-[6″-acetyl-2″-*O*-deoxyhexoside]-glucoside		+			
35	Flavonol	Kaempferol	+				+
36	Flavonol	Quercetin				+	
37	Flavonoid	Di-*O*-galloyl-glucoside *				+	
38	Isoflavan	Vestitol				+	
39	Prenyl flavonoid	Prenyl naringenin *	+			+	
40	Flavan-3-ol	(epi)Catechin-5-*O*-D-glycopyranoside *				+	
41	Flavanone	Liquiritigenin	+	+		+	
42	Flavanone	Methyl-Liquiritigenin *	+			+	
43	Flavanone	Liquiritigenin dimethyl ester *			+	+	
44	Flavanone	Padmatin *				+	
45	Flavanone	Maackiaflavanone B				+	
46	Phenolic acid	Ferulic acid				+	
47	Stilbene	Resveratrol				+	
48	Stilbene	3-Hydroxyresveratrol [Piceatannol]		+		+	+
49	Stilbene	Piacetannol-3-*O*-glucoside [Quzhaqigan]		+			
50	Stilbenolignan	Maackolin				+	
51	Dimeric stilbene	Resveratrol-piceatannol *				+	
52	Dimeric stilbene	Scirpusin A				+	
53	Dimeric stilbene	Scirpusin B				+	
54	Hydroxycoumarin	Esculin *					+
		OTHERS					
55	Omega-5 fatty acid	Myristoleic acid [Cis-9-Tetradecanoic acid] *	+	+		+	
56	Pterocarpan	Medicarpin				+	
57	Omega-3 fatty acid	Stearidonic acid *		+			
58	Omega 3-fatty acid	Linolenic acid *				+	
59	Pterocarpan	Maakiain [Inermin]	+	+			
60	Omega 3-fatty acid	Hydroxy linolenic acid *				+	
61	Higher-molecular-weight carboxylic acid	Oxo-nonadecanoic acid		+			
62	Oxylipins	13-Trihydroxy-Octadecenoic acid *				+	
63	Unsaturated fatty acid	Pentacosenoic acid *				+	
64	Anabolic steroid	Vebonol *				+	
65	Triterpenic acid	Ursolic acid *				+	
66	Product of chlorophyll degradation	Pheophytin A *	+	+		+	+

* Chemical compounds identified for the first time in *M. amurensis.*

**Table 2 molecules-28-02026-t002:** DPPH scavenging activity of *M. amurensis* extracts.

Compound	DPPH Scavenging Effect
	IC_50_ µg, 20 min
Quercetin	3.05 ± 0.12
MeOH extraction	22.26 ± 1.95
EtOH extraction	38.35 ± 4.09
Static CO_2_ extraction	30.11 ± 3.10
CO_2_ extraction 100 Bar	5.13 ± 0.42
CO_2_ extraction 300 Bar	15.32 ± 2.02

Data are presented as the mean values ± SEM, n = 3.

## Data Availability

Not applicable.

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
