# Peer review of "Maackia amurensis Rupr. et Maxim.: Supercritical CO2 Extraction and Mass Spectrometric Characterization of Chemical Constituents"

_molecules, 2023, doi:10.3390/molecules28052026_

Round 1
Reviewer 1 Report
The manuscript "Maackia amurensis Rupr. et Maxim.: Supercritical CO2-extraction and Mass Spectrometric Characterization of Chemical Constituents" compared 5 extraction methods for the obtention of phenolic compound and other metabolites from M. amurensis. I praise the precise MS-based identification workflow based on careful inspection of the MS/MS spectra. However, there are some aspects that must be considered. Some of the listed polyphenols are described with too much precision, e.g., compound 25 was annotated as a 7-O-glucoside despite nor the position (7 vs others) or the nature of the sugar (glucose vs galattose) can be distinguished by the sole MS/MS. I would encourage the authors to revise those identifications in the appendix and text. Table 1 is difficult to read, and the results are difficult to understand. How come only 3 of the 66 of the identified compounds was identified is common to all five approaches? Some more discussion is needed. In general, I believe that the science behind the proposed manuscript is solid enough for publication, but revisions are needed.
Author Response
Dear Reviewer.
Thank you very much for your detailed work on the text and your time spent.
Yes, you are absolutely right about compound 25 (Odoratin 7-O-glucoside), but we dare to note that we define the compounds found as tentatively identified and refer to mass spectrometric data obtained by other authors.
We also noticed this characteristic about compound 25 and several others found both in Maackia amurensis and in Radix astragali: firstly, its are quite rare and, as we now know, have been identified in Astragalus. We mention references to researchers who have worked with Astragal in the bibliography and in the table. This more than strange; it turns out that the these two medicinal plants are some kind of distant relatives.
We urgently requested Radix astragali from the North Caucasus and are now identifying them and trying to find similarities. We want to devote the following article to the similarity of this plants. We also updated the article a little, significantly corrected it and made several new statements.

Reviewer 2 Report
The study of Razgonova et al. entitled “Maackia amurensis Rupr. et Maxim.: Supercritical CO2-extraction and Mass Spectrometric Characterization of Chemical Constituents” is interesting, well written and innovative. I did not find any methodological errors. However, there are several minor remarks listed below which should be addressed:
1. Introduction: The supercritical CO2 extraction is considered as “green” method. This should be emphasized in the introduction.
2. Line 110: The authors used EtOH as co-solvent. What was the reason of selecting EtOH? Please explain this.
3. Figure 2: Please explain the legend.
4. Lines 171-178: I suggest to provide the intensities of the MS peaks. Were any MS/MS libraries such as NIST used during the identification? If so, the authors should write about it in the methodological part.
5. Lines 204-210: There is a lot of valuable information in Table 1. However, the discussion is very short. I suggest to extend it. For instance, Pentacosenoic acid, Vebonol and Ursolic acid are characterized by a relatively low polarity and in my opinion, this is why they were not found in ethanolic and methanolic extracts. Perhaps the presentation of theoretical logP values in Table 1 (for instance from the PubChem database or other) would provide some interesting observations. Furthermore, there are many papers on the solubility of identified natural compounds in ethanol, methanol, water and supercritical CO2.
Author Response
Dear Reviewer.
- Thank you very much for your work on our manuscript and for your time. Yes, CO2-extraction is a purely green technology. We corrected this omission in the text and added an additional paragraph in the Introduction. It is highlighted in blue.
- Since we mostly work with food matrices, we always use ethanol as a co-solvent in supercritical extraction.
- Thank you very much for your comments. The legend is added and highlighted in blue.
- We used extensive mass spectrometry literature from other authors in identifying various compounds isolated from Maackia amurensis (all authors are listed in References and listed in tables). For identification of these compounds, this seems to us sufficient.
- We do a lot of supercritical extractions of various food and plant matrices and a variety of substances behave very differently in supercritical extraction. I still cannot catch the general trend, the results of certain extractions from plants matrices vary greatly. The only rule: supercritical extraction, other things being equal, is much richer in isolated compounds than other types of extraction. In particular, we successfully detected unsolicited acid in Rhododendron adamsii during supercritical extraction. And such a chemical compound as vebonol, during ethanol maceration, I found in extracts of colored wheat grown in Siberia..

Reviewer 3 Report
In the manuscript, the authors undertook the identification of bioactive compounds in Maackia amurensis after the use of selected extraction methods.
In my opinion, the manuscript in its current form is not suitable for publication in Molecules. The topic is not something new and innovative. The authors made only the identification of the compounds - without quantifying the individual phytochemicals. In my opinion, there is a lack of at least one determination of biological potential (antioxidant, anti-inflammatory, antimicrobial) to indicate dependencies and correlations.
I recommend the authors to extend the scope of research, enrich the results and discussions, and resubmit the paper to the journal.
Author Response
Dear Reviewer.
Thank you very much for your work on the article and your time spent.
I would like to note that Maackia amurensis is an endemic and relict plant and grows in the Far East in its natural habitat. Detailed mass spectrometric studies of Maackia have not been carried out in other scientific laboratories, meanwhile, it has already been proven that it has bright healing properties in liver failure and, when compared with other modern medicines, significantly wins in efficiency and softness of the effect. These studies are mentioned in our bibliography.
Further, we noticed that many of the chemical compounds isolated in Maackia are also found in the medicinal plant Radix astragali, which in itself is of great medicinal interest, since these two plants are quite far from each other. We urgently requested Radix astragali roots from the North Caucasus for a more detailed study and will definitely write a comparative article of these two medicinally valuable plants: Maackia and Astragalus.
I wold like to note that detailed studies of Maackia at the modern scientific level are at the beginning of the journey, and you are absolutely right that a lotto new research can be done, including discoveries.
We have slightly corrected and supplemented the article with additional inclusions.
Sincerely yours.
Round 2
Reviewer 1 Report
I still believe that more adequate nomenclature should be used, e.g., Odoratin 7-O-glucoside vs Odoratin O-hexoside. I believe that the second would be more appropriate. However, the author are just following a general trend of assuming more information than real when MS/MS spectra are inspected.
Author Response
Dear Reviewer.
Thank you very much for your work on our text. We took into account your completely correct comments and corrected errors in the text and inaccuracies in the text.
Reviewer 3 Report
I still maintain the same opinion as in the previous review.
Author Response
Dear Reviewer.
We have significantly changed the text, took into account all your comments and added a chapter about antioxidant activity of Maackia. We plan to carry out experiments on antibacterial as well. Their description will be presented in the next article.